# Identification of Bioactive Compounds in *Warburgia salutaris* Leaf Extracts and Their Pro-Apoptotic Effects on MCF-7 Breast Cancer Cells

**DOI:** 10.3390/ijms26168065

**Published:** 2025-08-20

**Authors:** Lebogang Valentia Monama, Daniel Lefa Tswaledi, Tshisikhawe Masala Hadzhi, Makgwale Sharon Mphahlele, Mopeledi Blandina Madihlaba, Matlou Phineas Mokgotho, Leshweni Jeremia Shai, Emelinah Hluphekile Mathe

**Affiliations:** 1Department of Biochemistry and Biotechnology, School of Science and Technology, Sefako Makgatho Health Sciences University, Pretoria 0204, South Africa; lefatswaledi@gmail.com (D.L.T.); htmasala@gmail.com (T.M.H.); madihlabam@gmail.com (M.B.M.); mokgotho1250@gmail.com (M.P.M.); 2Department of Biomedical Sciences, Tshwane University of Technology, Pretoria 0183, South Africa; makgwalesharon@yahoo.com (M.S.M.); shailj@tut.ac.za (L.J.S.)

**Keywords:** apoptosis, *Warburgia salutaris*, breast cancer, anticancer, cancer therapy

## Abstract

The apoptotic mechanism is complex and involves many pathways. Defects can occur at any time along these pathways, resulting in malignant cell transformation and resistance to anticancer drugs. Collective efforts have made great progress in the implementation of natural products in clinical use and in discovering new therapeutic opportunities. This study aimed to screen volatile compounds of *Warburgia salutaris* leaf extracts and investigate their pro-apoptotic effects on MCF-7 cells. The approach was mainly based on determining cell viability using MTT and scratch assays, and DNA synthesis and damage using BrdU and comet assays, respectively. DAPI/PI stains were used for morphological analysis and expression was determined by RT-PCR and human apoptotic proteome profiler. *Warburgia salutaris* extracts exhibited antiproliferative effects on MCF-7 cells in a time- and dose-dependent manner. Acetone and methanol extracts exhibited low IC_50_ at 24, 48 and 72 h. Furthermore, the scratch test revealed that MCF-7 does not metastasise when treated with IC_50_. Expression showed upregulation of pro-apoptotic proteins and executioner caspases. Taken together, these findings suggest that leaves can promote apoptosis through the intrinsic apoptotic pathway, as observed by upregulation of the Bax and caspase 3 proteins. This paper provides new insights into the mechanisms of action of *W. salutaris* leaf extracts in the development of anticancer drugs.

## 1. Introduction

Cancer is one of the prominent causes of death worldwide, and different clinical approaches have helped increase survival rates. The global burden of disease and cancer collaboration (GLOBCAN) and the National Cancer Institute (NCI) have reported 17.5 million new cancer cases, and they are expected to increase to 23.6 million in 2030, globally [1]. According to the South African Cancer Association (CANSA), about 115,000 people in South Africa are diagnosed with cancer each year [2]. The most prevalent types of cancer in men include prostate, colorectal, and lung cancer, while in women it is breast and cervical cancer [1,2]. Breast cancer is reported as one of the most prevalent cancers in women worldwide, and developed countries have the highest rates of occurrence [3,4,5]. Luo et al. documented breast cancer as the most frequently diagnosed cancer in women, and the second leading cause of cancer-related deaths in the world [4,6,7].

Apoptosis, or programmed cell death, is a mechanism by which cells undergo death to control cell proliferation or in response to DNA damage [8]. The apoptotic mechanism is complex and involves many pathways, as defects can occur at any time along these pathways, resulting in malignant cell transformation and resistance to anticancer drugs [9,10]. The understanding of apoptosis has provided the basis for novel targeted therapies that can induce death in cancer cells or sensitise them to established cytotoxic agents and radiation therapy.

Drugs used to treat cancers are often associated with several undesired adverse side effects. Collective efforts have made great progress in the implementation of natural products in clinical use and the discovery of new therapeutic opportunities [11,12]. Natural products, including medicinal plants, are widely used in traditional medicine around the world and may present as sources of new cost-effective drugs [3,5].

Natural compounds and their derivatives are a useful source of molecules that can be tested for anticancer properties. Among these compounds, quinones are ubiquitous in nature and occur in animals, plants, and microorganisms [11,13]. The secondary metabolites derived from plants have shown remarkable anticancer properties, and the search for other new compounds that can serve as drugs or lead molecules for the semi-synthesis of anticancer drugs is ongoing [14].

*Warburgia salutaris* (Bertol. F.) CHIOV is an African medicinal plant that is generally used in traditional medicine to treat and manage cancers, among other diseases [3]. Furthermore, it is used to treat bronchitis, ulcers and oral thrush [14]. Previous studies highlighted that 69% of anticancer agents are designed from natural products used in cancer therapies. The phytoconstituents of the selected medicinal plant, *Warburgia salutaris*, were identified and found to be cytotoxic against several cancer cell lines [3,11]. However, the induced cell death mode and its involved pathways have not previously been elucidated, which provide the scope for further studies. The reported antiproliferative activity [3] of *Warburgia salutaris* extracts phytoconstituents and isolated compounds (Isomukadiaal acetate) against cancer cells lines provides an adequate justification for the selection of the plant species for the current study. Therefore, this study is aimed at the identification of bioactive compounds in *Warburgia salutaris* leaf extracts and the evaluation of their pro-apoptotic effects on MCF-7 breast cancer cells.

## 2. Results

### 2.1. GC-MS Analysis

The results of the gas chromatography-mass spectrometry (GC-MS) chromatogram presented in Table 1 show the separated phytochemicals found in the dichloromethane: methanol (50/50) extract of the *Warbugia salutaris*, and the compounds were eluted between 1.65 and 35.64 min. Table 1 represents the identification and detected phytochemicals constitutes in the leaf extracts, which were further characterised according to retention time (RT), compound name, molecular weight (Mw), and peak area (%). The table only shows phytochemicals with a quality score of ≥70% and above, indicating a high-confidence match with the NIST17 mass spectral library database as the reference spectrum. A score below <70% indicates lower confidence, which can result in poor spectral matches or false positives. Meanwhile, Table 2 shows the 13 identified compounds with anticancer activity.

### 2.2. Cell Viability Analysis of Warburgia salutaris Leaf Extracts of Mcf-7 Cells

A cell viability analysis of *Warburgia salutaris* leaf extracts in human breast adenocarcinoma (MCF-7) was evaluated using the 3-[4,5-dimethylthiazol-2-yl]-2,5 2,5-diphenyl tetrazolium bromide (MTT) assay, and the graphs for each solvent are shown in Figure 1 for all the extracts following 24, 48, and 72 h of treatment. The MCF-7 cells were treated with *W. salutaris* hexane, acetone, ethyl acetate, methanol, and crude aqueous leaf extracts at different concentrations ranging from 0.625 mg/mL to 10 mg/mL, hydrogen peroxide (1% v/v) served as the positive control, while untreated cells were the negative control. As observed in the graphs (Figure 1), the acetone (Figure 1B), and methanolic (Figure 1D) extracts exhibited less cell viability as compared to hexane (Figure 1A), ethyl acetate (Figure 1C), and aqueous (Figure 1E) extracts. Furthermore, the acetone and methanol extracts exhibited the lowest IC_50_ values in 24 h, 48, and 72 h compared to all the extracts.

### 2.3. DNA Synthesis of Warburgia salutaris Leaf Extracts on MCF-7 Cells

The effects of *Warburgia salutaris* leaf extracts on DNA synthesis in MCF-7 were investigated using the 5-Bromo-2′-deoxyuridine (BrdU) assay, and the results are presented graphically in Figure 2. The MCF-7 cells were treated with *W. salutaris* hexane, acetone, ethyl acetate, methanol, and aqueous crude leaf extracts at different concentrations (0.625–10 mg/mL); doxorubicin hydrochloride served as the positive control and untreated cells served as the negative control. Hexane, ethyl acetate, and distilled water extracts, as observed, appeared to stimulate significantly higher DNA synthesis than the untreated controls. Meanwhile, the methanolic and acetone extracts resulted in significantly lower DNA synthesis than the untreated control. The levels of DNA synthesis in treated cells with the two extracts were comparable to those observed in cells treated with the positive control.

### 2.4. Antimetastatic Effects of Warburgia salutaris Leaf Extracts on MCF-7 Cells

A scratch assay was used to assess the antimetastatic effect of *W. salutaris* using the MCF-7 cells treated with *W. salutaris* acetone (2.5 mg/mL) and methanol (5 mg/mL) crude leaf extracts, with doxorubicin hydrochloride (2.5 mg/mL) being the positive control and untreated cells as the negative control. The results depicted below in Figure 3 are images from capturing wounds after 24, 48, and 72 h, and a graphical representation after calculating the wound area. As observed, the scratch remained uniform in treated cells after 24, 48, and 72 h, while in untreated cells it was slightly closed after 24 h and completely closed after 48 and 72 h of incubation.

### 2.5. Morphological Analysis of MCF-7 Cells Treated with W. salutaris

A morphological assessment of MCF-7 cells after the treatment with IC_50_ of *W. salutaris* acetone (2.5 mg/mL) and methanolic extracts (5 mg/mL) was performed. The images were then captured as shown in Figure 4. Doxorubicin hydrochloride (2.5 mg/mL) served as the positive control, while untreated cells served as the negative control. The MCF-7 cells were stained with an equal volume and concentration of both Propidium Iodide (PI) and 4′,6-Diamidino-2-phenylindole (DAPI), which are dyes that stain cells that are undergoing cell death, or they have already undergone cell death due to nuclear fragmentation and chromatin condensation. The cells were then viewed under Zoe fluorescence cell imager (Bio-Rad, Hercules, CA, USA), and those that underwent cell death were more fluorescent, and the positive control had the highest fluorescence, followed by the methanolic and acetone extracts. Moreover, all three treatments are also viewed in brightfield as there were morphological changes compared to the untreated cells, which suggests that indeed there is apoptosis that was induced by the treatment.

### 2.6. DNA Damage Analysis of MCF-7 Cells Treated with Warburgia salutaris Leaf Extracts

An assessment of the induction of DNA damage to confirm the changes in morphology is observed in Figure 5, following the treatment of MCF-7 cells with the IC_50_ of *W. salutaris* acetone (2.5 mg/mL) and methanolic (5 mg/mL) extracts using the comet assay. Subsequently, the results were captured as depicted in Figure 4. Cells that underwent DNA damage are observed by comet tails and comet heads. The positive control had quite a number of damaged DNA, observed as several comet ‘tails’, compared to acetone and methanolic extracts. Moreover, the acetone extract had more potency and resulted in DNA damage as compared to the methanolic extract. Untreated cells had no damaged DNA, as there were no comet ‘tails’ observed. This therefore justifies that all the treated cells underwent DNA damage while the untreated cells had no damaged DNA, as there are no comet ‘tails’ observed.

### 2.7. Gene Expression by Reverse Transcription PCR (RT-PCR)

Following DNA damage, gene expression was determined using RT-PCR to observe the expression of apoptotic genes (*Bax*, *Bcl-2*, and *Caspase 3*) after the treatment of MCF-7 cells. GAPDH served as the control gene. The results were analysed using gel electrophoresis and images captured in Figure 6, together with the graphical representation of the relative expression of all genes. From the results obtained, it is observed that the caspase 3 expression was enhanced by the pro-apoptotic gene *Bax*, which was upregulated by treatment with plant extracts and the positive control, while bcl-2 was downregulated.

### 2.8. Protein Expression by Human Apoptotic Profiler Assay

Following gene expression, protein expression was performed to validate the specific apoptotic pathway induced by *W. salutaris* leaf extracts. An analysis of apoptotic-related proteins was determined using the human apoptotic proteome profiler, which detects multiple apoptosis-related proteins in untreated and treated cell lysates, and the results are presented graphically in Figure 7. MCF-7 cells were treated with *W. salutaris* acetone and methanol extracts and doxorubicin hydrochloride. Figure 7 shows a graphical representation of expressed target proteins Bax, Bcl-2, pro-caspase 3, and cleaved caspase 3. From the results obtained, the Bax protein was upregulated while the Bcl-2 in controls and in cells treated with the acetone and methanolic extracts was downregulated. In untreated cells, Bax was downregulated while Bcl-2 was upregulated. Similarly, in control and acetone extract-treated cells, cleaved caspase 3 was upregulated. Meanwhile pro-caspase 3 was downregulated in MCF-7 cells. Moreover, there is an upregulation of cytochrome c in all treated cells except for the untreated.

## 3. Discussion

The chromatogram detected a total of 36 compounds which were above the quality score of <70% and identified by the reference spectrum (Table 1). A total of 13 were potentially identified that showed anticancer activity, which are D-Arabinitol (RT: 9.9) [27], 4H-Pyran-4-one [28], 2,3-dihydro-3,5-dihydroxy-6-methyl- (RT: 11.04) [29], 2-Methoxy-4-vinylphenol (RT: 13.48) [17], 3 (2H) -isoquinolinone [30], octahydro- (RT: 17.41) [31], Drim-7-en-11-ol (RT: 18.87) [32], Neophytadiene (RT: 19.67) [33], n-Hexadecanoic acid (Palmitic acid) (RT: 21.12) [34], Isolongifolol (RT: 21.85) [35], Drimenin (from Drim derivatives) (RT: 22.67) [36], Neral (Citral isomer) (RT: 25.10) [37], Bis(2-ethylhexyl) phthalate (RT: 26.04) [38], Cholesterol epoxide (RT: 31.64) [39], and Vitamin E (α-Tocopherol) (RT: 33.46) [40]. Each of the identified compounds had different biological activities. D-Arabinitol, 3 (2H) -isoquinolinone, octahydro, and Neophytadiene Neral were reported to have antiproliferative activity [15,18,20,23]. The compounds 4H-Pyran-4-one, 2,3-dihydro-3,5-dihydroxy-6-methyl-, and 2-Methoxy-4-vinylphenol n-Hexadecanoic acid had antitumor activity [16,17,21]. Drim-7-en-11-ol showed cytotoxicity in breast and cervical cancer cell lines [19,22]. Isolongifolol was reported to have suppression of metastasis in melanoma [16]. Cholesterol epoxide had antiangiogenic activity [25], while Vitamin E was reported to have chemo-preventive properties [26]. Bis(2-ethylhexyl) phthalate had some controversy, as some studies show anticancer effects at low doses (1.2–10 µg/mL) [24]. Most of these phytochemicals identified in the GC chromatogram are terpenes that represent a vast and structurally diverse class of compounds exploited in the drug development for cancer and other diseases. The identification of these compounds in this study presents the cancer therapeutic properties of the leaf extracts of *Warbugia salutaris*.

Following extraction, crude extracts of *W. salutaris* were used to treat breast cancer cells to assess cell viability using the MTT assay. The results obtained are presented graphically in both a time- and dose-dependent manner to estimate the IC_50_ at 24, 48, and 72 h for each extract. From the findings, the extracts of *W. salutaris* leaves showed cytotoxic effects on MCF-7 cells both in a time- and dose-dependent manner, as observed by a decrease in cell viability. However, there were instances where cell viability increased after 48 and 72 h of treatment, suggesting that the molecules or compounds that are responsible for stopping cancer cell proliferation are depleted or lost activity with incubation time [3]. This may also mean that active compounds may not kill cancer but rather slow their growth and are one of the hallmarks of cancer cells, as they are resistant to chemotherapeutic drugs [41,42,43,44]. Hexane, ethyl acetate, and aqueous extracts had the highest IC_50_ values compared to the acetone and methanolic extracts of *W. salutaris* leaves. Generally, a higher value of IC_50_ is an indication of resistance of cells to the drug (plant extracts). Based on the observed IC_50_ values, the acetone and methanol extracts appeared to extract the highest concentration of active ingredients, hence the low IC_50_ values. The extracts of the methanolic and acetone extracts of *W. salutaris* were reported to contain drimane sesquiterpenes and Mukaadial and confirmed that these compounds possess antiproliferative activity against cancer cells (MCF-7, MDA-MB 231, and RMG-1 cells) [3,28]. Based on the findings of these studies, some of the antiproliferative activity observed in the current study can be attributed to these compounds [5,42].

In the current study, it was observed that the hexane, ethyl acetate, and aqueous extracts did not inhibit DNA synthesis, as evidenced by the percentage of thymine incorporation comparable to that of untreated control cells. The methanol extract resulted in a 50% DNA synthesis, while the acetone extracts, despite inhibiting DNA synthesis between 20 and 50% lower than the untreated control, were less active than the methanol extract. The findings suggest that *W. salutaris* leaf extracts exert their mechanism by inhibiting or rather slowing down DNA synthesis in MCF-7 cells by hindering DNA replication. Plants are a source of several bioactive compounds that affect the cell cycle and apoptosis and modulate various signal pathways as such; *W. salutaris* extracts are also a source, as observed by their ability to inhibit DNA synthesis in MCF-7 cells. Research [43] has shown that quercetin, which is also a powerful flavonoid, has several beneficial properties, such as anticancer activity. It has been shown to inhibit cell proliferation and induce cell cycle arrest and apoptosis in HeLa cells. Furthermore, the effect of phytoestrogens, flavonoids, and isoflavonoids on breast cancer cells was reported and found that in MCF-7 cells, when treated with compounds of 20–90 pM, they inhibited DNA synthesis by 50% [44,45,46].

Furthermore, MCF-7 cells were treated with acetone and methanolic extracts, and the wound was observed every 24 h for 3 days. Acetone, methanol extracts, as well as doxorubicin hydrochloride inhibited cell migration after 24, 48, and 72 h of incubation, while the wound created in untreated control cells closed within the first 24 h of incubation. These results suggest that the acetone and methanolic extracts do not promote the migration or proliferation of breast cancer cells. From the calculated wound area and the graph presented, it was observed that the wound area of the negative control decreased with time from 0 to 72 h. Compared to treated cells, the wound area of all treated cells remained intact, followed by a slight decrease in the wound area in cells treated with the acetone extract in 72 h. The bioactive compounds present in the acetone extracts may be rendered inactive with prolonged incubation. To ensure sustained antimetastatic activity of the extract, constant cell exposure must be maintained. Sultan et al. [47] reported the wound healing effects of methanolic extracts of the medicinal plant *Artemisia absinthium* L., which have been shown to have bioactive compounds such as flavone, octadecanoic acid, and flavonoid, among others, and the plant showed better wound healing properties [47,48,49]. Furthermore, research by Mabasa et al. showed that the antimetastatic effects of the n-butanol fraction of *R. communis*, which was extracted from the acetone solvent, inhibited the migration of MCF-7 cells when treated with sublethal concentrations of 0.1 and 0.2 mg/mL [48]. The plant showed bioactive compounds such as alkaloids, flavonoids, terpenes, saponins, and phenolics, which are also found in *W. salutaris* leaves [50].

Different apoptotic morphological characteristics, such as membrane blebs, apoptotic body formation, and/or dead cells, were observed in MCF-7 cells after treatment with extracts and doxorubicin. The methanolic extract and the positive control had more dead cells compared to the acetone extract, as indicated by less fluorescence. Meanwhile, the negative control did not have any dead cells, as there was less fluorescence and no morphological changes observed. Cells that undergo apoptosis or necrosis are characterised by different morphological features. As such, in late apoptosis and necrotic cells, the integrity of the plasma membrane and nuclear membrane is compromised, allowing the PI stain to pass through the membrane, intercalating into nucleic acids, and a red fluorescence is observed. Similarly, a DAPI stain can only pass inefficiently through an intact membrane and therefore bind to A-T rich regions of damaged DNA, which is observed by a blue fluorescence [49,51,52]. Serala et al. and Kntayya et al. reported that apoptosis is characterised by biochemical and morphological features, such as membrane blebbing, nuclear fragmentation, chromatin condensation, formation of apoptotic bodies, and loss of membrane permeability [45,53]. Ashok and Babu reported that *Vigna radiata*’s acetone and methanol extracts caused changes in apoptosis in MCF-7 cells [13]. In addition, the study suggests that the presence of secondary metabolites, including acids, alkaloids, flavonoids, phenols, quinones, tannins, terpenoids, and triterpenoids, is responsible for apoptotic changes [13,53,54,55]. *Warburgia salutaris* represent morphological changes similar to those of *Vigna radiata*, as they present the same secondary metabolites as reported by Khumalo et al., with similar secondary metabolites as reported in *Warburgia salutaris* [56].

To confirm the induction of apoptosis via genotoxicity, the comet assay was used. Comets originating from damaged cells have a distinct head, with a tail known as the head and tail. The heads and tails of treated and untreated cells were compared, and it was observed that the cells treated with acetone and methanol extracts and doxorubicin had the comet heads and tails characteristics of DNA damage. Moreover, the acetone treated cells appeared to have more comet heads and tails, and it is more fluorescent than other treated cells. Therefore, this suggests that extracts of *W. salutaris* induce apoptosis in MCF-7 cells through DNA damage. This evidence in the current study suggests that *Warburgia salutaris* leaf extracts might contain mannitol, a powerful OH scavenger that can, together with flavonoids, induce DNA damage in cells exposed to *Warburgia salutaris*, as the presence of flavonoids is reported [57,58]. Moreover, Somboro et al. have reported the detection of apoptosis using Annexin V and PI stains, in which the *W. salutaris* aqueous extract induced late apoptosis (necrosis) in HepG2 treated cells [59]. However, DNA damage is not direct, as it is caused by mitochondrial or membrane damage that results in extensive DNA fragmentation. This, therefore, is in support of the findings in the current study in that *Warburgia salutaris* could induce genotoxicity, which will therefore lead to apoptosis in MCF-7 cells.

Three genes related to apoptosis, namely *bax*, *bcl-2*, and *caspase 3*, were targeted at the mRNA level. From the results obtained, there was an increase in the expression levels of the *bax* gene in cells treated with the acetone and methanol extracts. The observed upregulation of the *bax* gene signals the induction of apoptosis. This notion is supported by the downregulation of the anti-apoptotic gene, *bcl-2*, in cells treated with *W. salutaris* leaf extracts [9,60,61,62]. In contrast to the negative control, the *bax* gene was downregulated while the *bcl*-2 gene was upregulated, indicating that apoptosis was not initiated in the untreated MCF-7 cells. However, the confirmation of cell death by apoptosis requires an extensive investigation on various apoptosis parameters, including but not limited to the analysis of *bcl-2* and *bax*, as well as *caspase 3*. There was an upregulation of *caspase 3* in cells treated with the extracts as well as doxorubicin, an indication and confirmation of apoptotic cell death. Taken together, *Warburgia salutaris* leaves promote programmed cell death (apoptosis), as seen by the increased expression of the *bax* gene and *caspase 3* gene, and the suppressed expression of the *bcl-2* gene. From these findings, *W. salutaris* leaf extracts notably induce the intrinsic apoptotic pathway, specifically the caspase 3-mediated apoptotic pathway, as observed by the upregulation of the *caspase 3* gene in all the treated cells.

For the investigation of pro-apoptotic proteins in MCF-7 cells treated with *Warburgia salutaris* leaves, a human apoptotic profiler was used to assess the levels of 35 apoptotic-related proteins. For this study, the focus is on the expression of pro-apoptotic proteins in comparison with the expression of anti-apoptotic proteins. From the observations, Bax and Bad proteins were upregulated in cells treated with either doxorubicin hydrochloride, acetone extract, or methanol extract, while downregulation of the Bax and Bad proteins was observed in the negative control cells. From the results, the Bcl-2 protein in the negative control is upregulated, whereas there is downregulation in all the treated cells, confirming the results obtained through RT-PCR. Pro-apoptotic proteins promote apoptosis by allowing the mitochondrial release of cytochrome c, while the opposite is true for anti-apoptotic proteins [9,62]. There was an upregulation of cytochrome c in cells treated with the extracts or doxorubicin. This may suggest that cytochrome c was released, and its release allowed the release of proteins such as SMAC/DIABLO and HTRA2/Omi, which function to block survivin, cIAP1/2 XIAP1, and XIAP2 [8,9,60]. Hence, in all the treated cells there was an upregulation of HTRA2/Omi and downregulation of survivin and XIAP. In the untreated cells, there was a downregulation of HtrA2/omi and Smac/DIABLO, while survivin and XIAP were downregulated. Furthermore, the release of cytochrome c can lead to the formation of homooligomeric protein complexes, known as the apoptosome, that alter the permeability of the mitochondrial membrane [8,60,61,62]. As such, once the apoptosome is activated, the cell may lose the ability to produce MOMP and therefore activate the caspase cascade [8,9]. From the results obtained, there is an upregulation of both pro-caspase 3 and cleaved caspase 3 in all the treated cells. The pro-caspase 3 is the inactive form of caspase 3, while cleaved caspase 3 is the active form of caspase 3. Furthermore, the conversion of pro-caspase 3 to cleaved caspase 3 serves as a crucial node of apoptosis, and once activated, there is no way to undo induced cell death. Wong highlighted that the loss of caspase 3 function and expression leads to breast cancer cell survival [8]. In the current study, one interesting aspect that emerged from the analysis is that there is little expression of caspases 3 in the negative control membrane; thus, no cytochrome c activity was observed. From the findings, it can be confirmed that *W. salutaris* leaf extracts induce apoptosis via the intrinsic apoptotic pathway, also known as the mitochondrial pathway. This was observed by the upregulated pro-apoptotic protein Bax compared to the expression of Bcl-2 that is downregulated; the upregulated cytochrome c and proteins involved the formation of the apoptosome and the activation of caspase 3, which subsequently will lead to apoptosis. Moreover, a significant increase in the Bax protein was observed, as opposed to the extrinsic apoptotic death receptor proteins. As such, this confirms that the *Warburgia salutaris* leaf extracts induce apoptosis through the intrinsic (mitochondrial) pathway.

## 4. Materials and Methods

The *Warburgia salutaris* plant material was collected from SANBI in Nelspruit, Mpumalanga. The plant specimens were then authenticated by a taxonomist in the Department of Biology and Environmental Sciences at the Sefako Makgatho Health Sciences University. The plant leaves were air dried, ground, and stored. About 50 g of ground plant leaves were extracted in 500 mL of different solvents, namely hexane (C_6_H_14_), acetone (C_3_H_6_O), ethyl acetate (C_4_H_8_O_2_), methanol (CH_3_OH), and distilled water. The extracts were then filtered using Whatman No 1 filter paper (Whatman No1, (Boeco Qualitative filter, grade 3 hw 125 mm, Hamburg, Germany) and concentrated at room temperature. The water extracts were freeze-dried using a vacuum freeze dryer (Vacutec, Roodepoort, Johannesburg, South Africa). Before conducting assays, all extracts were dissolved in dimethyl sulfoxide (DMSO) to yield a stock solution of 100 mg/mL and stored at 4 °C until use.

### 4.1. Gas Chromatography-Mass Spectrometry Analysis

GC-MS analysis was performed on the dichloromethane: methanol extract (50:50) of *Warbugia salutaris* using an Agilent 7890B gas chromatograph attached with an Agilent 5977B mass selective detector (MSD) (Agilent Technologies, Santa Clara, CA, USA). Separation of the mixture was conducted using an HP-5MS capillary column (30 m × 0.25 mm ID × 0.25 μm film thickness, stationary phase of 5% phenyl-methylpolysiloxane). An electron ionisation (El) system was selected and operated at 70 eV for detection. Helium gas with a high purification (99.999%) was used as a carrier gas at a constant flow rate of 1.0 mL/min with an injection of 1 μL sample volume, which was injected in the spitless mode, and the probe injector temperature was maintained at 290 °C, and the ion source temperature was set at 230 °C. The oven temperature was set at 50 °C and held for 2 min (isothermal), then increased to 300 °C at a constant rate of 10 °C/min, which was kept for every 10 min. This yielded a total running time of 35 min. The mass spectra were recorded in a full-scan mode with a 50–600 m/z scan range and a 0.3 s scan interval. The composition of the relative percentage of each component was calculated and compared between the total peak area and the average peak area using the ChemStation software (Version 4.03.016). The compound identification was analysed using the NIST17 mass spectral library database software (Version 3). The spectral library requires a minimum matching score quality of 70% and above for identification.

### 4.2. Cell Culture

MCF-7 cells were donated by Tshwane University of Technology (Acardia Campus), Department of Biomedical Sciences. The MCF-7 cells were cultured in an RPMI-1640 medium supplemented with 10% (*v*/*v*) foetal bovine serum (FBS) and 1% (*v*/*v*) antibiotic cocktail solution (10,000 unit/mL penicillin and 0.1 g/L streptomycin) in a humidified atmosphere of 5% and 37 °C. The cell culture medium was changed every 2–3 days and cells were seeded or sub-cultured for subsequent experiments once they had reached an 80–90% confluence.

#### 4.2.1. Cell Viability Analysis

A cell viability assessment was performed using the MTT assay described by van Meerloo et al. [63]. Cells were seeded in 96-well plates to allow cell adhesion. After 24 h, 100 μL of crude *W. salutaris* leaf extracts were added at different concentrations (from 0.625 mg/mL to 10 mg/mL), with hydrogen peroxide (H_2_O_2_) as a positive control and a negative control consisting of the medium and cells. The plates were then incubated for 24, 48, and 72 h with treatment at 37 °C and 5% CO_2_. Subsequently, 20 µL of 5 mg/mL of MTT reagent was added to each well and the plates were incubated for 4 h in a CO_2_ incubator and 37 °C. Following a 4 h incubation, 10 µL of DMSO was added into each well to solubilize the cells and dissolve the colour substance. For complete solubilisation, the plates were vigorously agitated for 1 min at room temperature, and then the optical density at 650 nm was read using a plate reader (SpectraMax iD3 MultiMode Microplate Reader Molecular Devices (United Scientific, Cape Town, South Africa). The percentage viability was calculated using the following formula:$$\mathrm{Pecentage}\;\mathrm{viability}\;\left(\%\right)=\frac{\mathrm{Absorbance}\;\mathrm{of}\;\mathrm{treated}\;\mathrm{cells}}{\mathrm{Aborbance}\;\mathrm{of}\;\mathrm{untreated}\;\mathrm{cells}}\;\times \;100$$

The 50% growth inhibition concentrations (IC_50_) of the extracts were then calculated from response curves.

#### 4.2.2. DNA Synthesis Analysis Using BrdU Assay Kit

The evaluation of DNA synthesis of *Warburgia salutaris* leaf extracts in MCF-7 cells was performed using the BrdU assay kit (Novus Biologicals, Centennial, CO, USA) following the manufacturer’s protocol. Cells were seeded in a 96-well plate at a density of 2 × 10^5^ cells/mL and incubated overnight at 37 °C and 5% CO_2_ to allow them to attach. Following incubation, the growth media was removed and replaced with the *Warburgia salutaris* leaf extract treatment with varying concentrations ranging from 0.625 mg/mL to 10 mg/mL, followed by incubation of the plates at 37 °C and 5% CO_2_ for 24 h. Before the end of the incubation period, a 20 µL 1× BrdU label was added to all wells, except for negative control wells. After the incubation period, the media were aspirated from all wells, 200 µL of the fixing/denaturing solution was added to all the wells, and the plates were incubated for 30 min at room temperature. The fixing solution was aspirated, and the plates were washed 3 times with 1x wash buffer. Then, 100 µL of Peroxidase Goat anti-Mouse IgG conjugate was added, followed by incubation of the plates at room temperature for 1 h. The Peroxidase Goat anti-Mouse IgG conjugate was aspirated, and the plate was washed 3 times with wash buffer, flooded with distilled water, and allowed to dry at room temperature. About 200 µL of TMB Peroxidase substrate was added to all the wells, and the plates were incubated for 30 min in the dark. Then 100 µL of the stop solution was added, and the plate was read at a wavelength of 450 nm using a microtiter plate spectrophotometer (SpectraMax iD3 multimode microplate reader, Molecular Devices (United Scientific, Cape Town, South Africa). The percentage viability was calculated using the following formula:$$\mathrm{Pecentage}\;\mathrm{viability}\;\left(\%\right)=\frac{\mathrm{Absorbance}\;\mathrm{of}\;\mathrm{treated}\;\mathrm{cells}}{\mathrm{Aborbance}\;\mathrm{of}\;\mathrm{untreated}\;\mathrm{cells}}\;\times \;100\;$$

#### 4.2.3. Antimetastatic Analysis Using Scratch Assay

The scratch assay was used to determine the antimetastatic effects of *W. salutaris* leaves on MCF-7 cells following a method described by Mabasa et al. [48]. Cells at a density of 5 × 10^5^ cells/well were seeded in a 6-well plate and then incubated for 24 h at 37 °C and 5% CO_2_ to allow the cells to attach. After incubation, the media was removed, and scratch was created using a 200 μL sterile pipette tip in each well. The cell images were captured in each well. Subsequently, cells were treated with methanol extract (5 mg/mL), acetone extract (2.5 mg/mL), and doxorubicin hydrochloride (2.5 mg/mL) as the positive control, with cells and media only as the negative control. Then, the cells were incubated for 24 h at 37 °C and 5% CO_2_. After 24, 48, and 72 h of incubation, images were captured to check the scratch and therefore calculate the wound sizes using the following formula:$$\mathrm{Closed}\;\mathrm{wound}\;\mathrm{area}\;\left(\%\right)=\frac{\mathrm{open}\;\mathrm{wound}\;\mathrm{area}\;\mathrm{at}\;\mathrm{specified}\;\mathrm{time}}{\mathrm{open}\;\mathrm{wound}\;\mathrm{area}\;\mathrm{at}\;0\;h}\;\times \;100\;$$

#### 4.2.4. Apoptotic Morphological Analysis Using DAPI/PI Staining

The apoptotic morphological analysis of MCF-7 cells treated with *Warburgia salutaris* leaf extracts was determined using DAPI/PI double staining, as described by Mou et al. [14], with amendments. Cells (1 × 10^6^ cells/flask) were seeded in a T25 flask and incubated overnight at 37 °C and a 5% CO_2_ atmosphere to allow cells to attach. After incubation, the medium was removed from all the flasks and replaced with a fresh medium containing either methanol extract (5 mg/mL), acetone extract (2.5 mg/mL), or doxorubicin hydrochloride (concentration), with cells and media only as the negative controls. The treated cells were incubated at 37 °C and a 5% CO_2_ atmosphere for 24 h. After incubation, the medium was removed, and washed cells were washed with sterile 1× PBS. Thereafter, 300 µL of 4% paraformaldehyde was added, and the flasks were wrapped and incubated for 1 h at 4 °C in a refrigerator. After incubation, 200 µL of Triton-X100 was added to all flasks, which were incubated in the dark at room temperature for 30 min. After 30 min of incubation, 100 µL DAPI/PI (1:1) dye was added, and the flasks were incubated at room temperature for 20 min. Following the incubation time, pictures were captured using the Zoe imager (Bio-Rad, Hercules, CA, USA). The excitation wavelength was 493 nm and the emission 636 nm for the PI stain, and the excitation for DAPI was 358 nm, and its emission was at 461 nm.

#### 4.2.5. DNA Damage Analysis by Comet Assay

Following the manufacturer’s protocol, the DNA damage of MCF-7 cells treated with *Warburgia salutaris* leaf extracts was determined using the comet assay kit (Abcam, Amsterdam, The Netherlands). MCF-7 cells were seeded in a 6-well plate at a density of 1 × 10^5^ cells/well and incubated overnight to allow them to attach. After 24 h, the growth medium was removed, and the cells were treated with acetone and methanolic extracts at 2.5 mg/mL and 5 mg/mL, respectively, and then incubated for 24 h at 37 °C and 5% CO_2_. Following incubation, the slides were viewed under fluorescence microscopy, and the images were captured. The excitation wavelength was 491 nm and the emission was 516 nm.

#### 4.2.6. Gene Expression by Polymerase Chain Reaction (PCR)

##### RNA Isolation

Total RNA was extracted from treated MCF-7 cells using the PureLink^®^ RNA mini kit (Invitrogen, Waltham, MA, USA) following the manufacturer’s protocol. Following cell culture at 37 °C for 24 h and treatment with IC_50_ of the *W. salutaris* leaf extracts, cells were harvested using a scrapper, collected into Eppendorf tubes, and centrifuged for 5 min at 12,000 rpm. The supernatant was discarded and the pellet rinsed with sterile PBS. The pellet was resuspended in 600 µL of lysis buffer supplemented with β-mercaptoethanol. Following resuspension, 360 µL of ethanol (100%) was added to the cell lysate and mixed by pipetting up and down. Then 700 µL of lysate was transferred to the GeneJET RNA purification column inserted into a collection tube, followed by centrifugation for 1 min at 14,000 rpm. The flow was discarded, and the purification column was placed back into the collection tube. The remaining lysate was centrifuged again at 14,000 rpm for 1 min, only the flow-through was discarded and the collection tube was placed back to the GeneJET RNA purification column then added 700 µL of the wash buffer. About, 700 µL of wash buffer 1 was added into the GeneJET RNA purification column and centrifuged for 1 min at 12,000 rpm. The flow-through was discarded, and the purification column was placed back into the collection tube. Then, 600 µL of wash buffer 2 was added to the GeneJET RNA purification column and centrifuged for 1 min at 12,000 rpm. The flow-through was discarded, and the purification tube was placed back into the collection tube. Wash buffer 2 (250 µL) was added to the GeneJET RNA purification column and then centrifuged for 2 min at 12,000 rpm. About 100 µL of nuclease-free water was added to the centre of the GeneJET RNA purification column to elute RNA, followed by centrifugation for 1 min at 12,000 rpm. The RNA was quantified using a nanodrop (Bio-Rad, Hercules, CA, USA) and used immediately for cDNA synthesis.

##### First Strand cDNA Synthesis

Complementary DNA (cDNA) was synthesised from each RNA sample (0.1 ng–5 µg) using a Revert Aid First Strand cDNA synthesis kit, following the manufacturer’s instructions (Thermo Scientific, Midrand, South Africa). Each reaction mixture contained components in the proportions highlighted in Table 3.

The reaction mixtures were gently mixed and incubated for an hour at 55 °C. After 1 h of incubation, the temperature was adjusted to 85 °C for an hour. The samples were then stored at −20 °C, pending a polymerase chain reaction.

##### Polymerase Chain Reaction (PCR)

In a 100 µL PCR tube, 2 µL of cDNA (corresponding to 5 µg of the total RNA input), 1 µL of the forward primer and 1 µL of reverse primer and 8.5 µL of nuclease-free water and 12.5 µL of the master PCR mix were mixed by pipetting. The following programme was set up in a thermal cycler: initial denaturation at 95 °C for 10 min, 30 cycles of denaturation at 95 °C for 30 s, primer annealing at 52 °C for 30 s, and an extension at 72 °C for 60 s, followed by a final extension at 72 °C for 7 min and the final hold at 4 °C. The PCR products were analysed on a 2% agarose gel electrophoresis containing 0.5 µg/mL ethidium bromide and then visualised and captured using a Chemi-doc image analyser (Bio-Rad, Hercules, CA, USA). The primer sequences are listed in Table 4 below.

#### 4.2.7. Protein Expression by Human Apoptotic Proteome Profiler

Protein expression was performed using a human apoptotic proteome profiler following the manufacturer’s protocol (R&D systems, Biotechne, Minneapolis, MN, USA). MCF-7 cells were seeded in TC25 flasks at a density of 1 × 10^6^ cells/flask and incubated overnight as previously described. After overnight incubation, the growth medium was removed and cells were treated with 2.5 mg/mL acetone and 5 mg/mL methanolic extracts of *Warburgia salutaris* leaves for 24 h, with doxorubicin hydrochloride as the positive control and untreated cells as the negative control. Cells were washed with PBS and then harvested using a scraper, transferred to a clean centrifuge tube, and centrifuged at 3000× *g* for 4 min. The supernatant was then discarded, and the pellet was washed with PBS. The pellet was resuspended in a lysis buffer and the lysate rocked at 4 °C for 30 min and centrifuged at 13,000× *g* for 5 min. The supernatant was transferred to a clean tube. Array buffer 1 of about 2000 µL was added to each well of the 4-well multi-dish, and, using tweezers, the arrays were inserted into each well with the array numbers facing upward. The plates were closed with a lid and incubated for an hour on a shaker on a rocking platform. During the blocking of the rows, 250 µL of each cell lysate was added to a tube that contained 1.25 mL of 1 row of buffer 1 to make a total volume of 1.5 mL for each cell lysate. After an hour of incubation, array buffer 1 was aspirated, and 1.5 mL of the mixture was pipetted into a 4-well multi-dish plate and incubated overnight on a rocking platform shaker at 4 °C. After incubation, each plate was removed from the plate and placed in its individual plastic, which contained 20 mL of 1x wash buffer. The lysate on the plate was aspirated, the plate was rinsed with distilled water and air dried. The arrays were then washed with 1× wash buffer for 10 min for a total of 3 washes. After the wash step, 0.015 mL of the reconstituted detection antibody cocktail was diluted into 1.5 mL of 1× array buffer 2/3 and aspirated 1.5 mL into each well, then the arrays were placed in each individual well, and the plate was incubated on a rocking platform shaker for an hour. After incubation, plates were removed from the 4-well multi-dish, the detection antibody cocktail was aspirated, and the plate was washed with distilled water and allowed to dry. The top plastic sheet protector was carefully removed and placed on a lab absorbent towel on top of the membrane to cover the remaining visualisation agent mixture. The membrane was placed with the identification numbers facing an autoradiography film cassette. The membranes were exposed to X-ray film for 3 min before visualisation using a Chemi-doc image analyser (Bio-Rad).

## 5. Conclusions

The chromatogram of the GC-MS analysis performed resulted in the high-confidence identification of 13 anticancer properties within the *Warbugia salutaris* leaf extracts, which signifies its traditional usage as a traditional medicine. The results depicted that *Warburgia salutaris* extracts inhibit cell proliferation and DNA synthesis of breast cancer cells. Moreover, *W. salutaris* induces DNA damage, which then induces cell death as observed by the morphological changes in breast cancer cells. The treated cells underwent cell death via the intrinsic apoptotic pathway, as observed by the upregulation of the *Bax* gene and downregulation of the *Bcl-2* genes. Furthermore, the *Caspase 3* activation further highlighted the execution stage of apoptosis (caspase 3-mediated pathway). These results broaden the understanding of *Warburgia salutaris* leaves in drug development since the leaves possess various anticancer compounds such as Palmitic acid, 2-Methoxy-4-vinylphenol, and Cholesterol epoxide that also have the potential to induce apoptosis, which makes the plant a promising cancer therapy. Therefore, the findings of the current study suggest that the leaves of *Warburgia salutaris* possess a phytoconstituent with pro-apoptotic inducing properties in MCF-7 cancer cells through the caspase 3-mediated pathway.

## Figures and Tables

**Figure 1 ijms-26-08065-f001:**
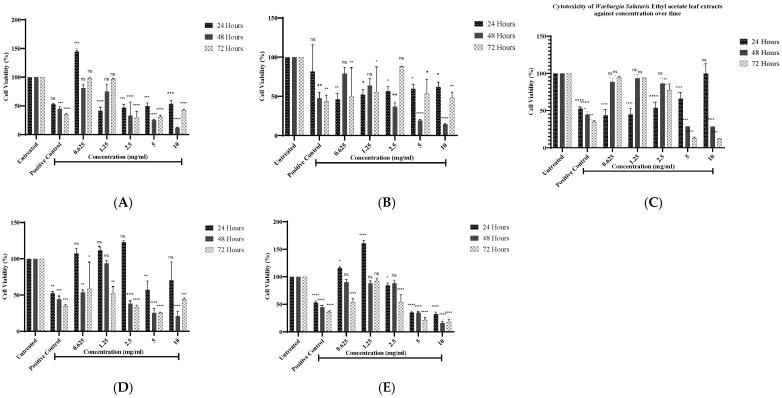
Cell viability analysis of hexane (**A**), acetone (**B**), ethyl acetate (**C**), methanol (**D**), and aqueous (**E**) from *W. Salutaris* leaf extracts on MCF-7 cell line using MTT assay in time- and dose-dependent; hydrogen peroxide served as positive control, and untreated cells served as the negative control. Data represent the mean ± Standard Deviation of two independent experiments and * *p* ≤ 0.05, ** *p* ≤ 0.01, *** *p* ≤ 0.001 and **** *p* ≤ 0.0001 indicate significant differences to the control group. ns, not significant.

**Figure 2 ijms-26-08065-f002:**
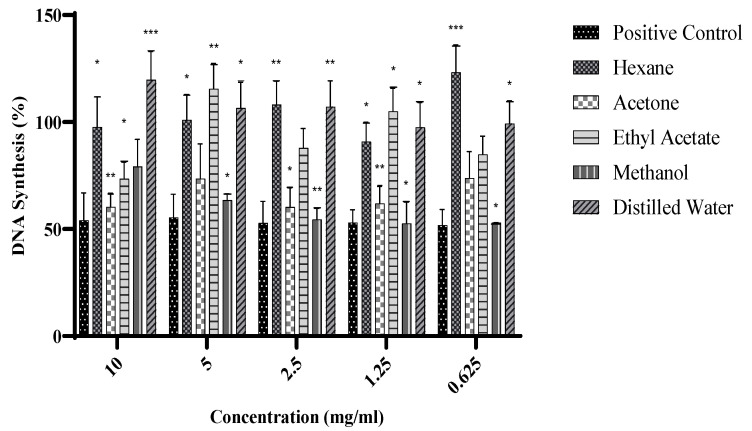
DNA synthesis of MCF-7 cells after exposure to *Warburgia salutaris* crude leaf extracts for 24 h. Doxorubicin hydrochloride served as positive control, and untreated cells served as the negative control. Data represent the mean ± Standard Deviation of two independent experiments and * *p* ≤ 0.05, ** *p* ≤ 0.01, and *** *p* ≤ 0.001 indicate significant differences from the Positive Control group.

**Figure 3 ijms-26-08065-f003:**
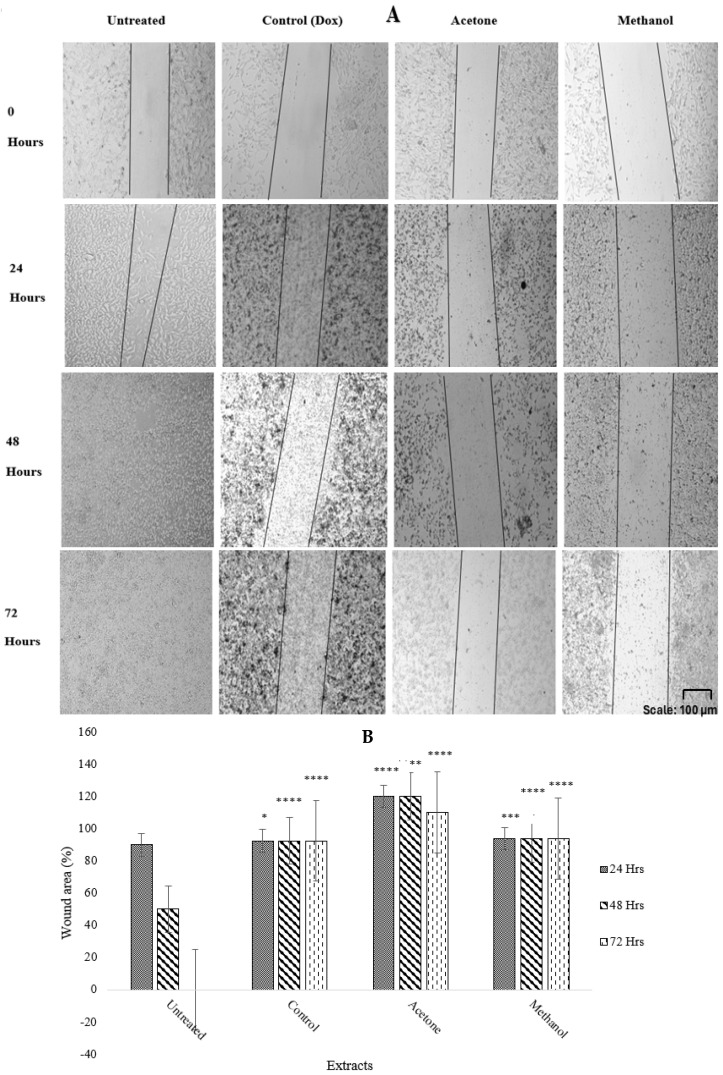
(**A**) Images of antimetastatic effects and (**B**) graphical representation of percentage wound area of *W. salutaris* acetone and methanolic leaf extracts on MCF-7 cells after 0, 24, 48, and 72 h of exposure. Doxorubicin hydrochloride served as positive control, and untreated cells served as the negative control. Data represent the mean ± Standard Deviation of the experiment and * *p* < 0.05, ** *p* < 0.01, *** *p* < 0.001, and **** *p* < 0.0001 indicate significant differences from the negative control (untreated).

**Figure 4 ijms-26-08065-f004:**
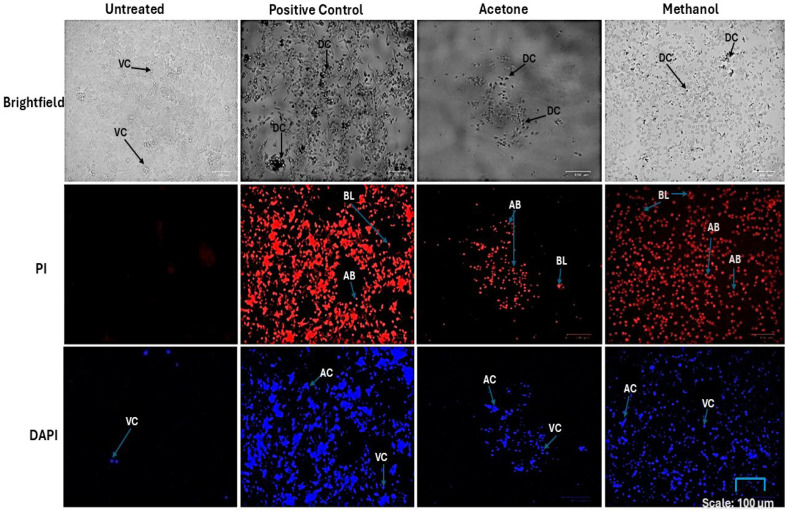
Images of PI and DAPI-stained MCF-7 cells post-24 h of exposure to treatment with acetone and methanolic *W. salutaris* leaf extracts. Doxorubicin hydrochloride served as the positive control and untreated cells served as the negative control. In the figure, DC (dead cells), AC (apoptotic cells), and VC (viable cells). BL (membrane blebbing), DC (dead cells), and AB (apoptotic bodies).

**Figure 5 ijms-26-08065-f005:**
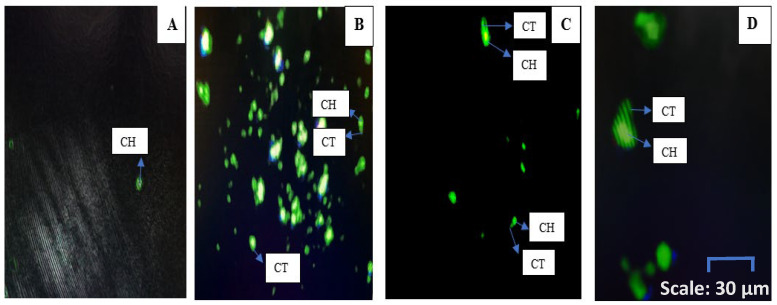
The effects of *W. salutaris* leaf extracts on DNA damage on MCF-7 cells after 24 h of treatment. (**A**) Untreated cells, (**B**) positive control is doxorubicin hydrochloride, (**C**) *W. salutaris* acetone extract, and (**D**) *W. salutaris* methanolic extract. In the figure, CH indicates the comet head, while CT indicates comet tails.

**Figure 6 ijms-26-08065-f006:**
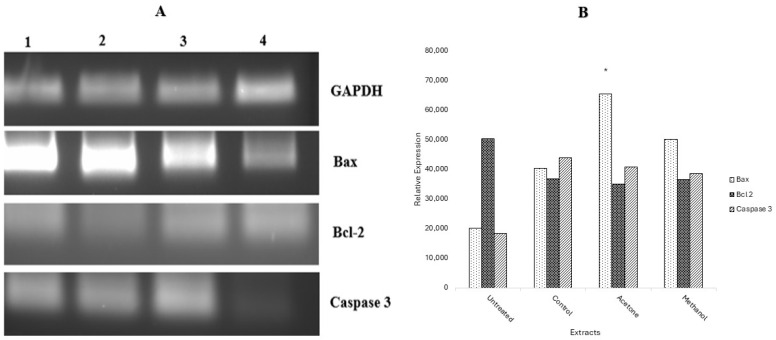
(**A**) Gel electrophoresis following RT-PCR analysis of the expression of apoptotic-related genes *bax*, *bcl-2*, *caspase 3*, and GAPDH. Lane 1: acetone extract; lane 2: methanolic extract; lane 3: doxorubicin hydrochloride; and lane 4: untreated cells. (**B**) Graphical representation of the expression levels of apoptotic-related genes for acetone and methanolic leaf extracts. Data represent the mean ± Standard Deviation of the experiment, and * *p* < 0.05 indicates significant differences from the negative control (untreated).

**Figure 7 ijms-26-08065-f007:**
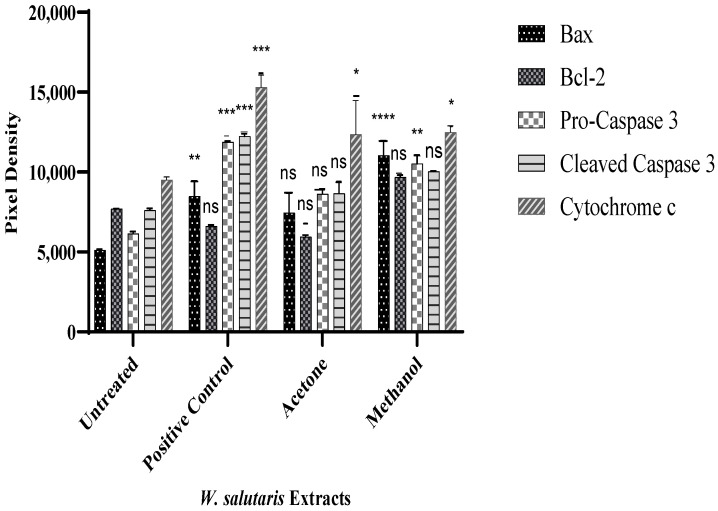
Pixel density of apoptotic proteins expressed by MCF-7 cells treated with *W. salutaris* leaf acetone and methanolic extracts. Doxorubicin hydrochloride served as the positive control while untreated cells served as the negative control. Data represent the mean ± Standard Deviation of the experiment and * *p* < 0.05, ** *p* < 0.01, *** *p* < 0.001, **** *p* < 0.0001, and ns, meaning not significant, indicate significant differences from the negative control (untreated).

**Table 1 ijms-26-08065-t001:** GC-MS phytochemical analysis of compounds identified in the leaf extract of *Warburgia salutaris* with a quality score of ≥70% identification. The three dots “…” indicate that there are compounds that are below 70% identification score that are omitted.

Peak No.	Retention Time (min)	Compound Name	Molecular Weight (g/mol)	Molecular Formula	Peak Area (%)
1	1.649	Carbon dioxide	44.009	CO_2_	0.36
2	2.072	Methylene chloride	84.93	CH_2_Cl_2_	22.40
3	3.670	1-Nitro-2-acetamido-1,2-dideoxy-D-glucitol	252.22	C_8_H_16_O_7_	0.08
4	4.004	N-Methoxy-N-methylacetamide	103.12	C_4_H_9_NO_2_	0.06
5	4.331	6,6,7,7,8,8,8-Heptafluoro-5,5-bis(trifluoromethyl)oct-1-ene	374.15	C_10_H_7_F_13_	0.06
6	5.197	2-Chloropent-4-ene carboxylic acid, methyl ester	148.6	C_6_H_9_ClO_2_	0.22
7	5.753	Dimethyl sulfoxide	78.13	C_2_H_6_OS	0.11
8	6.119	Cyclopentane, 2-ethylidene-1,1-dimethyl-	124.23	C_9_H_16_	0.15
9	6.549	Butanoic acid, 2,2-dimethyl-3-oxo-, methyl ester	130.14	C_6_H_10_O_3_	0.12
10	6.903	2-Pentanone, 4-hydroxy-	102.13	C_5_H_10_O_2_	0.19
11	7.494	Oxepine, 2,7-dimethyl-	122.17	C_8_H_10_O	0.13
12	7.820	2(5H)-Furanone, 5,5-dimethyl-	112.13	C_6_H_8_O_2_	0.07
13	8.342	Hexanoic acid, 1-methylethyl ester	158.24	C_9_H_18_O_2_	0.23
14	8.564	2-Cyclopropyl-2-propanol	100.16	C_6_H_12_O	0.11
15	8.994	Butanoic acid, 3-hydroxy-	104.11	C_4_H_8_O_3_	0.25
16	9.282	Hexanal dimethyl acetal	146.23	C_8_H_18_O_2_	0.41
17	9.858	D-Arabinitol	152.15	C_5_H_12_O_5_	0.64
18	10.289	Octadecyl propyl ether	312.58	C_21_H_44_O	0.29
19	10.537	4-Bromo-2-methylpent-2-enoic acid, methyl ester	193.04	C_6_H_9_BrO_2_	0.32
20	11.043	4H-Pyran-4-one, 2,3-dihydro-3,5-dihydroxy-6-methyl-	144.13	C_6_H_8_O_4_	0.39
…	…	…	…	…	…
41	21.117	n-Hexadecanoic acid (Palmitic acid)	256.43	C_16_H_32_O_2_	9.95
42	21.846	Isolongifolol	222.37	C_15_H_26_O	1.79
43	22.211	2,4-Pentanedione, 3-(phenylmethyl)-	190.24	C_12_H_14_O_2_	3.34
44	22.665	Cyclopropane, 1,1-dimethyl-2-(2-methyl-1-propenyl)-	124.22	C_9_H_16_	4.31
45	23.266	Cycloisolongifolene, 9,10-dehydro-	202.33	C_15_H_22_	2.26
46	23.592	2,4-Heptadiene, 2,6-dimethyl-	124.23	C_9_H_16_	1.83
47	24.104	Cyclohexane, 3,3,5-trimethyl-	126.24	C_9_H_18_	1.05
48	24.503	2-Dodecen-1-yl (-) succinic anhydride	266.38	C_16_H_26_O_3_	1.10
49	25.051	Neral	152.24	C_10_H_16_O	1.68
50	25.731	Hexadecanoic acid, 2-hydroxy-1-(hydroxymethyl)ethyl ester	330.51	C_19_H_38_O_4_	1.18
51	26.040	Bis(2-ethylhexyl) phthalate	390.56	C_24_H_38_O_4_	0.68
52	26.319	Docosane	310.61	C_22_H_46_	0.59
53	26.684	4-(2,2-Dimethyl-6-methylenecyclohexyl) butanal	194.32	C_13_H_22_O	0.88
54	27.195	Heptacosane	380.75	C_27_H_56_	0.51
55	27.475	(E)-15,16-Dinorlabda-8(17),11-dien-13-one	288.47	C_20_H_32_O	0.49
…	…	…	…	…	…
70	35.637	Campesterol	400.07	C_28_H_48_O	0.05

**Table 2 ijms-26-08065-t002:** GC-MS phytochemical analysis of compounds with the potential of anticancer, anti-tumour, and antiproliferative activity identified in the leaf extract of *Warbugia salutaris*.

Peak No.	Retention Time (min)	Compound Name	Molecular Weight	Molecular Formula	Peak Area (%)	Reported Activity	References
**17**	9.858	D-Arabinitol	152.15	C_5_H_12_O_5_	0.64	Anti-proliferative (via oxidative stress induction in cancer cells)	[15]
**20**	11.043	4H-Pyran-4-one, 2,3-dihydro-3,5-dihydroxy-6-methyl-	144.13	C_6_H_8_O_4_	0.39	Cytotoxic to tumour cells (analogues inhibit cell cycle progression)	[16]
**26**	13.476	2-Methoxy-4-vinylphenol	150.18	C_9_H_10_O_2_	1.35	Antitumor (induces apoptosis in leukaemia cells)	[17]
**34**	17.409	3(2H)-Isoquinolinone, octahydro-	153.23	C_9_H_15_NO	4.37	Antiproliferative (binds to DNA in cancer cells)	[18]
**37**	18.869	Drim-7-en-11-ol	222.37	C_15_H_26_O	2.40	Cytotoxic to breast cancer cells (MCF-7)	[19]
**39**	19.667	Neophytadiene	278.05	C_20_H_38_	3.10	Anti-proliferative (induces apoptosis in colon cancer)	[20]
**41**	21.117	n-Hexadecanoic acid (Palmitic acid)	256.43	C_16_H_32_O_2_	9.95	Anti-tumour (modulates lipid metabolism in cancer cells)	[21]
**42**	21.846	Isolongifolol	222.37	C_15_H_26_O	1.79	Suppresses metastasis in melanoma	[16]
**44**	22.665	Drimenin (from Drim-derivatives)	234.33	C_15_H_22_O_2_	4.31	Potent cytotoxic activity against HeLa cells	[22]
**49**	25.051	Neral (Citral isomer)	152.24	C_10_H_16_O	1.68	Anti-proliferative (via ROS generation)	[23]
**51**	26.040	Bis(2-ethylhexyl) phthalate	390.56	C_24_H_38_O_4_	0.68	Controversial (some studies show anticancer effects at low doses)	[24]
**62**	31.635	Cholesterol epoxide	402.66	C_27_H_46_O_2_	0.21	Anti-angiogenic (blocks tumour vascularization)	[25]
**66**	33.458	Vitamin E (α-Tocopherol)	430.72	C_29_H_50_O_2_	0.30	Chemo preventive (reduces oxidative stress in tumours)	[26]

**Table 3 ijms-26-08065-t003:** cDNA synthesis reaction mixture (proportion of components added).

Template	2 µL
Random Hexamer Primer	1 µL
Nuclease-free Water	9 µL
5× Reaction Buffer	4 µL
RiboLock RNase Inhibitor	1 µL
10 mM dNTP	2 µL
RevertAid M-MuLV RT (200 U/µL)	1 µL
Total Volume	20 µL

**Table 4 ijms-26-08065-t004:** Shows forward and reverse primer sequences used to amplify *Caspase 3*, *Bax*, *Bcl-2*, and *GAPDH* [63,64,65,66].

Gene	Primer Sequences
*Bax*	Forward: 5’TCCCCCCAGAGGTCTTTT 3′ Reverse: 5’ CGGCCCCAGTTGAAGTTG 3′
*Bcl-2*	Forward: 5’CTGCACCTGACGCCCTTCACC 3′ Reverse: 5’CACATGACCCCACCGAACTCAAAGA 3′
*Caspase-3*	Forward: 5’CCATGGGTAGCAGCCTCCTTC 3′ Reverse: 5’TGCGCTGCTCTGCCTTCT 3′
*GAPDH*	Forward: 5’TGCGCTGCTGCTCTGCCTTCT 3′ Reverse: 5’CCATGGGTAGCAGCTCCTTC 3′

## Data Availability

No new data were created or analyzed in this study. Data sharing is not applicable to this article.

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
