# Peer review of "Identification of Bioactive Compounds in *Warburgia salutaris* Leaf Extracts and Their Pro-Apoptotic Effects on MCF-7 Breast Cancer Cells"

_ijms, 2025, doi:10.3390/ijms26168065_

Round 1

Reviewer 1 Report

Comments and Suggestions for Authors

The present study explores the effects of Warburgia salutaris leaf extracts on MCF-7 human breast cancer cells. Multiple assays were used to evaluate various cellular responses: MTT and scratch assays assessed cell viability and migration; BrdU and comet assays measured DNA synthesis and damage, respectively; DAPI/PI staining was used for morphological analysis; and RT-PCR determined gene expression profiles. The extracts inhibited cell migration and induced the expression of proapoptotic genes, particularly Bax and caspase-3. These results suggest that W. salutaris promotes apoptosis through the intrinsic mitochondrial pathway and may have therapeutic potential for the treatment of breast cancer.

Recommendations for Authors

1. Clarify the primary objective of the study in the introduction. The distinction between the general relevance of natural compounds and the specific hypothesis tested in MCF-7 cells should be explicitly established.

2. Strengthen the rationale for plant selection. Include a detailed justification based on ethnobotanical records, pharmacological data, or previous scientific studies demonstrating the antitumor potential of W. salutaris.

3.Improve data visualization and figure quality:

  • Figures 1 and 3 are too small and difficult to interpret.
  • Standardize or normalize the data in Figure 1 to facilitate direct comparisons.
  • Improve the resolution and clarity of Figures 4 and 5.
  • Include additional tables or graphical summaries to improve data presentation and interpretation.

4. Provide detailed information on extract standardization. Explain how extract batch variability was controlled to ensure consistency and reproducibility across experiments.

5. Expand and strengthen the discussion. Compare the findings with the existing literature and contextualize the results within the broader framework of cancer biology and phytotherapy.

6. When appropriate, include a discussion at the molecular level to deepen the mechanistic understanding of the observed cellular responses.

Comments on the Quality of English Language

The manuscript is generally understandable; however, numerous grammatical, syntactical, and stylistic errors hinder readability and scientific clarity. Specific problems include:

Inconsistent use of verb tenses (e.g., inappropriate switching between the past and present tense).

Subject-verb agreement errors.

Unnatural sentence constructions.

A thorough linguistic review by a native English-speaking editor with experience in scientific writing is strongly recommended to improve clarity and professionalism.

Author Response

Comment 1: Clarify the main objective of the study in the introduction. The distinction between the general relevance of natural compounds and the specific hypothesis tested in MCF-7 cells must be explicitly established.

Response 1: We appreciate the suggestion to clarify the main objective of the study. The aim of the study was incorporated into the introduction section of our manuscript in lines 77-79 in the last paragraph stated as follows; Therefore, this study is aimed at the identification of bioactive compounds in Warburgia salutaris leaf extracts and evaluation of their pro-apoptotic effects on MCF-7 breast cancer cells. The specific hypothesis was therefore addressed in conclusion paragraph spanning paragraph lines 587-589 stated as follows: Therefore, the findings of the current study suggest that the leaves of Warburgia salutaris possess phytoconstituent with pro-apoptotic inducing properties in MCF-7 cancer cells through the caspase 3 mediated pathway.

Comment 2: Reinforce the justification of plant selection. Include a detailed justification based on ethnobotanical records, pharmacological data or previous scientific studies that demonstrate the anti-tumour potential of W. salutaris.

Response 2: The introduction section of our manuscript provides detailed justification for the selection of W. salutaris, including its ethnobotanical records in paragraph lines 67-70 which reads; Furthermore, its ethnobotanical records its usage in the treatment of bronchitis, ulcers, and oral thrush [14]. Previous study highlighted that 69% of anticancer agents are designed from natural products used in cancer therapies. Phytoconstituents of the selected medicinal plant, Warburgia salutaris, were identified and found to be cytotoxic against several cancer cell lines [3, 11]. However, the induced cell death mode and its involved pathways have not been previously elucidated, which provide scope for further studies.  Meanwhile, the pharmacological data or previous scientific studies that demonstrate its anticancer potential are addressed in paragraph lines 74-79.

Comment 3: Improve data visualisation and figure quality: Figures 1 and 3 are too small and difficult to interpret. Standardise or standardise Figure 1 data to facilitate direct comparisons. Improve the resolution and clarity of Figures 4 and 5. Include additional tables or graphical summaries to improve data presentation and interpretation.

Response 3: We have revised figures (figures 1 and 3) to improve clarity, resolution, and readability.

Comment 4: Provide detailed information on the standardisation of extracts. Explain how the variability of the extract batches was controlled to ensure consistency and reproducibility in the experiments.

Response 4: Extracts were reconstituted in 0.1% DMSO for standardization or working solutions. Single extract batch was used throughout the study. Justification for variability was recommended for further studies.

Comment 5: Broaden and strengthen the debate. Compare the findings with the existing literature and contextualise the results in the broader framework of cancer biology and phytotherapy.

Response 5: Our discussion section thoroughly compares our findings to existing literature and contextualizes the results in the wider context of cancer biology and plant therapy. Furthermore, it is worth noting that our study contributes novel insights by investigating  DNA Synthesis using BrdU assay, gene and protein expressions by RT-PCR and Human apoptotic proteome profiler, respectively. DNA Damage by comet assay as they have not been previously explored in the medicinal plants’ context.

Reviewer 2 Report

Comments and Suggestions for Authors

The manuscript describes the identification of bioactive compounds in Warburgia salutaris leaf extracts and their pro-apoptotic effects on MCF-7 breast cancer cells. The authors used a range of experimental methods including GC-MS analysis, MTT assays, BrdU assays, scratch assays, comet assays, morphological staining, and RT-PCR to characterize the extract's phytochemicals and assess their impact on cell viability, DNA synthesis, metastasis, DNA damage, and apoptotic gene expression. The study is well-structured and methodologically sound, with detailed experimental design and comprehensive data presentation. However, the manuscript has some flaws, several improvements must be performed:

Major Comments:

  1. The manuscript lacks any results on the cytotoxicity of the leaf extracts on non-cancer cells (e.g., normal breast epithelial cells). Measuring selectivity is needed before concluding therapeutic relevance.
  2. The PCR data suggest apoptosis induction via the intrinsic pathway, further validation—e.g., protein-level proof (e.g., Western blotting for Bax, Bcl-2, cleaved caspase-3)—would strengthen the conclusion.
  3. The study would benefit from including an Annexin V/PI or TUNEL flow cytometry assay quantitatively distinguish between early and late apoptosis.
  4. The manuscript contains occasional grammatical and syntactic errors. A thorough English language revision by a native speaker or professional editor is recommended.

Minor Comments:

  1. Some figure legends are too brief or lack essential experimental details. Ensure each figure legend is self-contained.
  2. Concentrations are inconsistently reported (e.g., mg/mL vs µg/mL). Standardize throughout.
  3. Some abbreviations are defined multiple times or not at all. Ensure all abbreviations are defined on first use only.

The study is promising and could contribute valuable insights into plant-based anticancer therapies, it requires additional data (especially on selectivity and apoptosis quantification) and clearer presentation before it can be considered for publication.

Author Response

Major Comments

Comment 1:The manuscript lacks any results on the cytotoxicity of the leaf extracts on non-cancer cells (e.g., normal breast epithelial cells). Measuring selectivity is needed before concluding therapeutic relevance.

Response 1: We appreciate the proposal to assess the cytotoxicity of leaf extracts in non-cancer cells. Although we recognize the importance of evaluating selectiveness, the current research focuses on the anti-cancer effects of the extracts on cancer cell lines. Future studies will examine the effects of extracts on normal cells, to further clarify their therapeutic potential.

Comment 2: The PCR data suggest apoptosis induction via the intrinsic pathway, further validation—e.g., protein-level proof (e.g., Western blotting for Bax, Bcl-2, cleaved caspase-3)—would strengthen the conclusion.

Response 2: We welcome the suggestion to validate our findings at the protein level. We have incorporated the results on  the expression of multiple apoptosis-related proteins using a human apoptotic proteome profiler.

Comment 3: The study would benefit from including an Annexin V/PI or TUNEL flow cytometry assay quantitatively distinguish between early and late apoptosis.

Response 3: We appreciate the reviewer’s suggestion to include Annexin V/PI or TUNEL flow cytometry. In our study, we used DAPI and PI dyes to assess apoptosis, which provided useful insight into the apoptotic mechanism. Future studies will build on these findings and incorporate additional methods such as Annexin V/PI or TUNEL flow cytometry tests to further clarify the mechanisms of apoptosis and provide a more comprehensive understanding of the apoptosis process.

Comment 4: The manuscript contains occasional grammatical and syntactic errors. A thorough English language revision by a native speaker or professional editor is recommended.

Response 4: The grammatical and syntactic errors were addressed.

Minor Comments:

Comment 1: Some figure legends are too brief or lack essential experimental details. Ensure each figure legend is self-contained.

Response1: Figure legends are revised to make each legend self-contained

Comment 2: Concentrations are inconsistently reported (e.g., mg/mL vs µg/mL). Standardize throughout.

Response 2: Our concentration units  are in mg/ml, µg/mL concentration emanate from comparative previous studies. Nonetheless all the units were standardized. 

Comment 3: Some abbreviations are defined multiple times or not at all. Ensure all abbreviations are defined on first use only.

Response 3: Addressed.

Round 2

Reviewer 1 Report

Comments and Suggestions for Authors

Dear authors

Thank you for your diligent work and responsiveness in revising the manuscript. We appreciate the time and effort you invested in addressing the comments and suggestions. The improvements you have made have significantly enhanced the quality and clarity of your work.

Reviewer 2 Report

Comments and Suggestions for Authors

Authors have revised their manuscript according my suggestions. I suggest the manuscript can be considered to accept by the journal.